# Resettlement willingness: From a village environmental perspective

**Chengxiang Wang**[1,2¤a]*, **Pinrong He**[1¤a], **Chang Gyu Choi**[2¤b]

**1** Institute of Land and Urban–Rural Planning, Huaiyin Normal University, Huai'an, Jiangsu, China, **2** Urban Design Analysis Lab, Graduate School of Urban Studies, Hanyang University, Seoul, Republic of Korea

¤a Current address: Huaiyin Normal University, Huai'an, Jiangsu, China
¤b Current address: Graduate School of Urban Studies, Hanyang University, Seoul, Republic of Korea
* liutanghe@hanyang.ac.kr

**Data Availability Statement:** All relevant data are within the manuscript and its Supporting information files.

**Funding:** This research was supported by the National Natural Science Foundation of China

## Abstract

Breaking the limitations of the urban perspective, there is an urgent need to study the influence of the village environment on the willingness of rural households to resettle. This paper explored the determinants and the mechanism of village environment factors on resettlement willingness using full-sample survey data (872,414 households) of 1382 administrative villages in Huai'an, a typical agricultural area in Eastern China. The result revealed that environmental factors generally have a greater impact on the spatial heterogeneity of resettlement willingness, in the order of natural environment, economic environment, social environment, and policy environment; among which geographic location, housing conditions, behavioral tendency of farmers and planning guidance are the key factors. In addition, the absolute location of the urban area in the geographic region has a significantly greater effect than that of the county, and the "following behavior" of the farmers affected their resettlement decision. Therefore, differentiated policies should be formulated according to the spatial distribution of the resettlement willingness, building a dual-core village and town system within the county.

## Introduction

Flow, mobility and relocation have become the commonplace in the trend towards "global cities". The attraction from the cities has led to a rapid depopulation of the countryside, resulting in the rural–urban resettlement of farmers [1, 2]. Dramatic changes in rural living space and land structure [3, 4], and the prevalence of phenomena such as "hollow villages" [5] and "elderly villages" [6, 7] have attracted widespread attention from governments, scholars, and the public regarding the social problems caused by these issues. In China, the sudden reform and opening-up triggered explosive urbanization and thus a mega rural–urban migration, bringing 740 million peasants to the cities (1978–2021) [8]. In response to rural decay, China has begun to centralize rural settlements on a piecemeal basis, but China's rural villages are very different, and the degree of willingness to relocate from villages varies significantly [9, 10]. However, China's top-down system of governance, as well as its hierarchical town system, dictates that China's settlement policies focus on cities and towns, thus leading to a number of

(52078237); and the National Research Foundation of Korea grant, which is funded by the Government of South Korea (NRF-2020R1A2C1008509). The funders played an important role in the study. The first grant leader, Chengxiang Wang, was responsible for the conceptualization, methodology, validation, data curation, writing, etc. Chang Gyu Choi, the leader of the second fund, was responsible for conceptualization, resources, supervision, project administration, etc. in this study. A detailed description of this is given in the author contribution section of our paper.

**Competing interests:** The authors have declared that no competing interests exist.

policy and practice deficits. The outline of the 14th Five-Year Plan for National Economic and Social Development and the 2035 Vision Goals of the People's Republic of China also clearly put forward that "the strategy of new urbanization centered on human beings should be promoted in depth, and a greater number of people can enjoy a higher-quality urban life." [11]. Therefore, to realize a people-centered rural reconstruction, the willingness of farmers must be respected more. Presently, it is necessary to shift the research perspective from urban to rural areas and analyze the impact of the rural environment on the willingness of farmers to relocate from the bottom up, so as to effectively protect the rights and interests of farmers and rationally carry out relocation work.

Willingness to resettle–the classical demographic, sociological and geographic research topic–has been widely studied. From the perspective of population migration, the classical push–pull theory better explains the dynamic mechanism of the migration process of agricultural households, and the differences in the natural, social, and economic environments between the place of origin and the migration destination are an important reason for population migration [12–14]. Some scholars have used the theories of neoclassical economics [15], new economic migration, and labor market segmentation [16] to try to identify the multidimensional influences on the decision of farm households to migrate to cities, but the focus of attention has mainly been on the pull of the city [17]. Among these factors, "pull" factors, such as high income [12], better housing conditions [18], more employment opportunities [19], improved education and medical resources [20, 21], a higher level of government governance [19], and a fairer political atmosphere in cities [22], have been widely recognized by scholars. However, a top-down research perspective can lead to outdated results, and the localized factors of villages and farmers cannot be ignored. In fact, the "push" factors of the villages in the emigration area, such as the level of agriculture [23], employment opportunities [19], medical and educational resources [24], and the ecological environment [25], have seldom attracted the attention and research of scholars [26, 27].

The willingness of households to resettle is a prerequisite for the ability of the population to migrate. In recent years, Chinese academics have conducted a great deal of localized research on the characteristics and influencing factors of farmers' willingness to resettle. Existing research shows not only that the willingness of agricultural households to resettle has obvious differentiated characteristics in different groups, different times, and different regions [13], but also that the willingness of agricultural households to resettle depends on the destination, as they tend to be more inclined toward relocating to a county or small town near to where they are domiciled [28], and their willingness to enter large cities to resettle is lower. At the same time, farmers' willingness is affected by many factors, consisting mainly of the living environment of the original village and town, terrain, transportation convenience, compensation intensity, supporting policies, regional economic level, urban–rural connection and other external conditions, as well as personal characteristics of farmers such as age and education level [29–31]. Also playing a role are family characteristics, such as income level and structure, employment structure, total family population, number of people in family homestead, area and scale of contracted land, renovation time of the current house, area and structure of the house, location and orientation of the house, survival relationships between the family and the place of origin, etc [10, 17, 32–35]. However, existing studies are mostly obtained through questionnaires or interviews, with most of them using sampling and dynamic survey data, while lacking full-sample data. This style of data means that most of these studies take the individual or household as the basic unit, and the research methods mostly comprise simple percentage statistics, binary logistic regression analysis, factor analysis, relational conceptual models, probit models, and so on, and fewer research methods relating to spatial scales are used. Recognizing and analyzing willingness to resettle on the spatial scale from the

perspective of geography is more helpful for obtaining a systematic understanding of the mechanism of interaction between a willingness to resettle and the geographic environment, helps to deepen understanding of the basic theories of population geography, and promotes the exploration of the scientific frontiers of population geography.

To summarize, the following aspects of current research need improvement: 1) there are only a few studies that analyze the mechanisms of how features of the countryside influence the willingness to resettle from the perspective of the countryside and seek a way for developing the countryside. 2) Existing research on the willingness of farmers to resettle lacks analysis of full-sample data, and this lack reveals a certain degree of one-sidedness in the characteristics of the law. 3) When there is a lack of analysis of the spatial differences in the factors influencing villagers' willingness to resettle compared to different influencing factors in different villages, it is difficult for proposed countermeasures to be tailored to the specific needs of individual villages. Based on the above shortcomings, this study aims to examine the law of willingness to resettle in villages of typical agricultural areas in eastern China from a geographic perspective, in order to make up for the lack of micro-spatial scales in the existing research on urban–rural population migration. This thesis proposes an analytical framework that analyzes the mechanisms of influence on willingness to resettle in four environmental dimensions: village nature, economy, society, and policy. On the basis of systematically analyzing the spatial distribution characteristics of village willingness to resettle, the spatial differentiation of determinants and the scale of influence are revealed for the first time by using the MGWR method. This study also empirically analyzes the influence mechanisms of the four dimensions of environmental factors, and finally puts forward relevant policy recommendations.

## Theoretical framework

### Farmers' resettle decision process and village environmental system

The resettlement decision-making process of Chinese farmers is divided into two stages: one is "wanting" to resettle, and the other is realizing resettlement. The former generates resettlement willingness, and the latter generates resettlement behavioral outcomes, which are mainly manifested in the form of purchasing or renting a house. Thus, willingness is the origin of behavior and is more influenced by the rural settlement. The basic driving force for the attraction of resettlement places is the difference between the place out of departure and the destination in terms of natural, economic and social development. Since the Chinese countryside is a communized society, people are not atomized one by one as in cities, but in villages as a group. When the willingness is accumulated to a certain level, a collective decision regarding village resettlement is formed. Therefore, the study of farmers' resettlement in villages needs to be carried out in a certain space rather than individually. Willingness to leave the village for resettlement is more influenced by the environmental factors of the village, and it is the overall response to the environment and policies affecting the farmers' resettlement decisions.

Therefore, we argue that villages' willingness to resettle is a product of the coupling of farmers and the rural environment, and that it is the four dimensions of the natural environment, the economic environment, the social environment, and the policy environment that together constitute the environmental determinants of the village [36]. The natural environment of the village characterizes the advantages and disadvantages of the natural conditions, geographic location, and ecological environment of the human living environment and reflects the relationship between the living space and the natural geographic environment. The village's economic environment characterizes the advantages and disadvantages of the village environment in terms of income, industrial development, regional circulation, living conditions, infrastructure conditions, etc., reflecting the relationship between the living space and

the village's economic environment. The village's socio-cultural environment characterizes the convergence of choices in the villages from which people move due to the "Shu Ren" (acquaintance) social network constructed by long-term blood and geographic ties, reflecting the relationship between the living space and the village social network environment [35]. The policy environment of the village system reflects the overall arrangement of the national and local governments in the region for the future development of the village, reflecting the relationship between the living space and policy guidance. The resettlement decision-making process is the re-siting of the living space caused by the uneven or insufficient development of the four elements in the process of village environment evolution, in which the willingness of farmers to live mainly in the village in the place of relocation is generated, which is more affected by the elements of the village environment.

## A theoretical framework for analyzing resettlement willingness

The degree of willingness to resettle in villages is the result of the joint influence of multidimensional factors, which is manifested in the diversity of influencing factors, the interactivity of the pathways of action, and the spatial heterogeneity of the effects of action. From the point of view of the impact effect (Fig 1), four types of elements, including the village's natural environment, economic environment, social environment, and policy environment, all have the potential to influence the generation and the level of willingness to resettle. From the perspective of the pathway of action, the generation of will is not dominated by a single element in most cases, but rather is the result of the joint action of multiple environmental elements. From the perspective of the spatial effect of the action, the degree of feedback to different environmental factors varies, resulting in spatial differences in the degree of influence of factors on farmers in different spatial locations.

The natural environment of a village, which is inherent to the natural evolution of a given area, is a prerequisite for the choice of living space by farming households and plays a fundamental role in the process of generating the will to resettle. The mechanisms by which elements of the natural environment affect the willingness to resettle are mainly manifested in three areas: natural endowment, geographical location, and land resources. First, the influence of natural endowments on village living space is mainly reflected in the natural conditions of the village, such as topography, river and lake water systems, and natural disasters [25, 37]. On the one hand, an undulating terrain and rich water system are conducive to the formation of

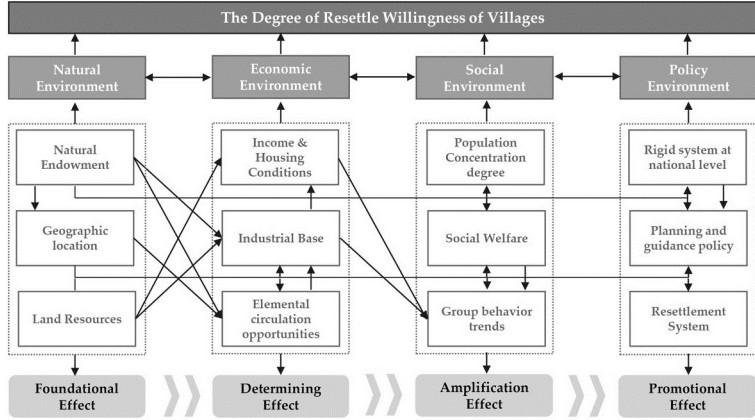

**Fig 1. A theoretical framework for the degree of resettle willingness of villages.**

unique natural landscapes, which is conducive to the creation of better rural landscapes, the promotion of rural tourism development, and the enhancement of the village economy, and at the same time enhance the willingness of farmers to return to their hometowns to spend their old age there, which in turn reduces the willingness of farmers to resettle. On the other hand, a rugged terrain and dense water network will also reduce the degree of village accessibility, affect the economic development of the village, and reduce the convenience of life, which will lead to the reduction in farmers' satisfaction with the village living space. In addition, natural disasters can increase the danger posed to the village living space and the uncertainty of village business income, thus increasing the willingness of farmers to resettle. Secondly, the geographical location of villages, influenced by objective factors such as natural barriers and administrative divisions, can result in problems such as a lack of public service facilities, high resource mobility costs, and fewer opportunities driven by outside development, leading to a loss of confidence among farmers in the development of the village living space, and thus triggering a willingness to resettle. Thirdly, land resources mainly include arable land and residential land, and there is a big gap between the land resources occupied by farmers in different regions. Chinese farmers have a strong "land plot"; if the arable land area is large and the soil cultivation conditions are good, farmers can obtain enough income by staying in villages and continuing to engage in agriculture and will not relocate if there is not a big gap between the conditions in other areas and those in cities [38]. If other conditions are not too different from those in a city, they will not resettle. On the other hand, the size of the homestead may produce very different effects in different regions [23]. In some regions with better transportation conditions, farmers may be able to substantially improve their living conditions through renovation because of the large homesteads, which may discourage them from resettling. In other regions, guided by local policies, a larger homestead means a higher replacement payment or better-quality housing for resettlement, which may result in a stronger willingness to resettle [39].

The economic environment of the village is the most direct manifestation of the economic conditions of farmers and plays a determining role in the process of generating farmers' willingness to resettle. The overall economic environment of a village is reflected in three aspects: income and housing conditions, the industrial base of the village, and elemental circulation opportunities. Above all, income and housing conditions are the most direct causes of a willingness to resettle; an increase in income increases the ability to buy a house in a town [28, 40], thus increasing the probability of the willingness to resettle. Generally speaking, the enhancement of housing conditions will reduce the probability of the willingness to resettle, but in some special areas (such as new industrial zones, ecological reserves, etc.), the standards of housing demolition and resettlement in the city are the same, and the large size of good-quality housing can mean receiving higher replacement payments, thus increasing the willingness of farmers to resettle. Furthermore, the industrial base of villages–including the history of industrial development, the diversity of industrial structures and the size of markets–is a prerequisite for the employment and development of farming households. Some rural areas have a weak industrial base and a single structure, with high opportunity costs for goods and services to enter the market, making it difficult to drive employment through industry, and surplus labor can only create a new living environment through resettlement [41]. Finally, elemental circulation opportunities provide a comprehensive characterization of the economic and transportation location [42]. Those areas far from economic, political, and transportation hubs have weak mobility of factors such as labor, capital, and technology, and thus poor market accessibility, which limits the number of livelihood options and consumption of farmers, forcing them to choose to resettle.

The social environment of the village has an amplifying effect on the willingness of farmers to resettle, and in the process of the long-term evolution of the rural settlement space, the

village organization forms more unified values and concepts, and this kind of social network of "Shu Ren (acquaintance)" has a greater impact on the willingness of farmers to resettle [36, 43]. This kind of impact is mainly reflected in the degree of population concentration, social welfare, and trends in group behavior. Firstly, the degree of population concentration includes population density and settlement size. A village settlement is a social network space formed by blood and geography, and the degree of population concentration represents, to a certain extent, the degree of connection of the social network; in general, in settlements with a high degree of connection, the willingness of farmers to resettle is lower [44, 45]. Secondly, social welfare is mainly reflected in the protection of education, medical, and other facilities. In areas where social service facilities are limited, the cost of obtaining social services for farmers is high, the supply of social welfare is insufficient, and it is difficult for the government to achieve absolute parity in public service facilities, meaning that farmers cannot easily obtain high-quality social services through their own development and have no choice but to resettle [21]. Thirdly, the trend of group behavior is the main the influencing factor on the existing behavior of the farmers in the village, their future behavior and willingness to resettle [46]. Due to blood relationships and mutual comparison in the village society, purchasing a house in a city has become a symbol of affluence or family status [28], and at the same time, resettling or purchasing a house in a town will destroy the social network relationship of the original village, meaning that the farmers will follow the trend and resettle blindly.

The village policy environment implicitly affects the willingness of farmers to resettle and plays a promotional role in facilitating the amplification or elimination of the willingness to resettle. The village's institutional policy environment influences the mechanism of willingness to resettle, which is mainly manifested in the rigid system at the national level, the planning and guiding policy of the local government, and the system of resettlement. Firstly, the rigid system at the national level mainly includes the land system and the ecological conservation system. China's homestead property rights system determines that after farmers have resettled, their original homes cannot be freely traded or mortgaged, making it difficult to effectively convert them into capital in the resettlement decision-making process. The ecological conservation system influences the willingness of farmers to resettle by designating ecological protection zones and restricting the development of basic services in these zones [24]. Secondly, the planning and guiding policy of the local government is mainly the local government's positioning of the future development of the village [47]. The "National New Urbanization Plan (2014–2020)" [48] also proposed "scientific guidance of rural housing and settlement construction"; during this period of time, all levels of government completed the full coverage of the planning of the layout of towns and villages. In the plans, villages are divided into key villages, characteristic villages, and general villages (this area should be labeled to indicate that the names of the divisions differ from place to place), and farmers are guided to gather in an orderly manner in key villages and characteristic villages, with farmers responding to this positioning in their resettle decision-making process. Thirdly, the resettlement policy consists of three main aspects: a subsidy policy, new agricultural insurance policy, and property rights policy. In regions where the resettlement policy has not yet been implemented, the decision of farmers to resettle will be influenced by the fact that the minimum subsistence guarantee, old-age insurance, and property rights of farmers cannot be guaranteed after resettlement [49].

## Data sources, methods and model construction

### Overview of the study area and data sources

We selected a rural area of Huai'an City in eastern China as our study area. Huai'an City is a municipal administrative district under the jurisdiction of Jiangsu Province, one of the most

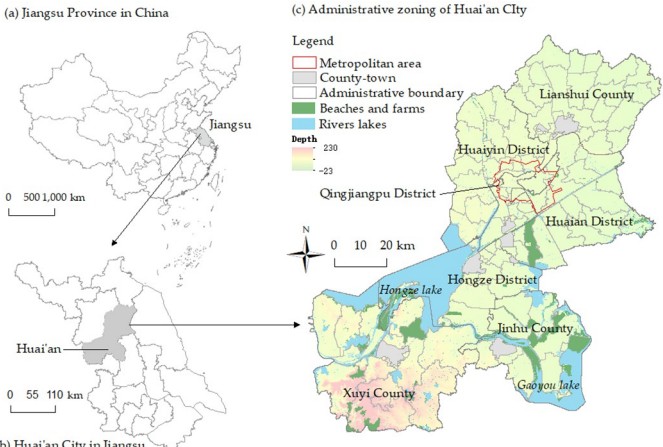

**Fig 2. (a) Jiangsu Province in China; (b) Huai'an City in Jiangsu; (c) Administrative zoning of Huai'an City.**

developed provinces in eastern China, and is located on the north–south demarcation line of Qinling–Huai River in China, with a total area of 10,030 square kilometers, of which 69.39% is covered by plains, and 11.39% is covered by water. The total grain output was 4,892,500 tons in 2022. The per capita grain output of 1,745.81 pounds, twice the national level, making this China's typical plains agricultural area, as shown in Fig 2. The administrative boundary data and elevation data are acquired from the Resources and Environment Science and Data Center of the Chinese Academy of Sciences (www.resdc.cn) [50]. As of the end of 2019, the municipal jurisdiction of the city totaled 95 townships and streets, and 1693 administrative villages or residential committees related to agriculture, with a rural population of 2,380,100 people, a permanent rural population of 1,804,000 people, and a rural resident population of 1,800,400 people.

The development of Jiangsu Province has always been unbalanced, with its long-term formation of three sub-regions with large gaps between southern, central and northern Jiangsu, and Huai'an is at the geographic center of northern Jiangsu. For a long time, Huai'an has been a net population outflow area, and the main city to which the population flows is Nanjing. Population loss has brought many problems to rural Huai'an. Rural Huai'an has experienced a significant loss of vitality, serious aging, and a lagging industrial development. Our data show that the rate of vacant houses in rural Huai'an is 17.13%. It is undeniable that with the implementation of a series of plans or policies to revitalize northern Jiangsu, Huai'an has experienced a certain amount of development, especially as the comprehensive strength of the city has increased, absorbing a certain amount of the local rural population. For example, Xuyi County and Gaogou Township, under the jurisdiction of Huai'an City, have a balanced population. However, Huai'an as a whole is still typically strong in the cities and weak in the countryside. In most areas of Huai'an, the countryside still does not have the strength to leave farmers behind, and resettlement is still the dominant way for farm households to urbanize. We believe that rural Huai'an is a typical area for studying the willingness of farm households to resettle.

## Variable selection and preparation

This study took the administrative village as the smallest statistical unit and used the group willingness rate of all farmers in the village instead of the individual willingness of farmers,

**Table 1. Selection of variables for the analytical model at the village level.**

| Factor | Variable | Description | VIF |
|---|---|---|---|
| Dependent variable | | | |
| Willingness to resettle at the village scale (WRV) | | Ratio of the number of households willing to resettle to the total number of households in the administrative village (%) | |
| Natural Environment Elements | | | |
| Natural Endowments | Water ($X_1$) | Proportion of water area to total area within a village (%) | 1.40 |
| | Terrain ($X_2$) | Average slope of the administrative village (°) | 1.17 |
| Geographical location | A11Y-City ($X_3$) | Travel time by car to the center of Huai'an City (min) | 2.31 |
| | A11Y-County ($X_4$) | Travel time by car to its county government site (min) | 3.32 |
| Land Resources | Farmland ($X_5$) | Ratio of cultivated area to number of people (acre) | 1.18 |
| | Homestead ($X_6$) | Average household size of homesteads (acre) | 1.47 |
| Economic Environment Elements | | | |
| Income and Housing Conditions | Revenue ($X_7$) | Average annual income of resident households (10,000 yuan) | 2.80 |
| | Building ($X_8$) | Ratio of number of buildings ($\geq$ 2 stories) to households (%) | 1.09 |
| | Q-Housing ($X_9$) | Proportion of houses with acceptable quality (%) | 1.30 |
| Industrial Base | GDP ($X_{10}$) | Annual fiscal revenue of its town (million yuan) | 1.16 |
| | Industry ($X_{11}$) | Number of industrial enterprises registered (pcs) | 1.12 |
| Elemental circulation opportunities | Network ($X_{12}$) | Ratio of road network equivalence factors[1] to village area (km/ km$^2$) | 1.54 |
| | External ($X_{13}$) | Travel time by car to the nearest transport station (min) | 1.65 |
| Social Environment Elements | | | |
| Population Concentration Degree | Decentralized ($X_{14}$) | Ratio of decentralized households (%) | 1.62 |
| | Density ($X_{15}$) | Ratio of permanently settled households to area (pcs/km$^2$) | 1.17 |
| | Household-Pop ($X_{16}$) | Population per household (pcs) | 1.38 |
| Social Welfare | High-school ($X_{17}$) | Travel time by car to the nearest high school (min) | 3.62 |
| | A11Y-town ($X_{18}$) | Travel time by electric bike to its town government site (min) | 1.45 |
| Group Behavioral Trends | Vacant ($X_{19}$) | Proportion of vacant (abandoned) buildings (%) | 1.50 |
| | Homebuyer ($X_{20}$) | Proportion of homebuyer in cities (%) | 1.94 |
| Policy Environment Elements | | | |
| Rigid policy | Restricted ($X_{21}$) | Proportion of natural villages with development restricted (%) | 1.06 |
| Planning Policy | Improve ($X_{22}$) | Proportion of natural villages with development policies (%) | 1.09 |

[1] Road network equivalence factors are converted according to speed: 0.375 for village roads, 0.5 for township roads, 0.625 for county roads, 0.75 for provincial roads, and 1 for national roads.

reflecting the relatively consistent willingness trend of farmers in the village micro-region constructed through the common social network. It analyzed the farmers' willingness to resettle as having been influenced by the factors related to the place of departure (the village) on the scale of the village. The dependent variable of the model was willingness to resettle at the village scale (WRV), which is the ratio of the number of households with willingness to resettle to the total number of households in the village, where "resettle" refers to the abandonment of the village house by farmers through inkind replacement or monetary replacement and excludes intra-village resettlement.

Based on the theoretical analytical framework of the mechanism of influence wrought by village environmental elements on the willingness to resettle constructed in the previous section, and taking into account the scientific nature, comprehensiveness, consistency of the evaluation scale (village scale), and the availability of data, the eleven factors listed in Table 1 were

selected, and twenty-two indicators were chosen to char-acterize the village environmental elements.

## Research methods

We used Moran's *I* to explore the spatial distribution characteristics of WRV. Global Moran's I indicator was used to determine whether there is spatial autocorrelation of WRV in the study area, and then to determine whether our study was statistically significant. The local autocorrelation (in this paper, we use hotspot analysis in local autocorrelation) was used to detect the local spatial clustering of WRV and the type of clustering, in which the high-value clustering areas were called "hot spot areas" and the low-value clustering areas were called "cold spot areas" [28].

We first performed the analysis using OLS regression methods. OLS regression is a traditional linear regression analysis method that is very widely used for impact factor analysis. However, OLS only estimates parameter correlations at the global level and does not take into account the effect of spatial variation, which does not truly reflect the impact effect. Fotheringham et al. proposed that geographically weighted regression (GWR) is a commonly used econometric local regression model to account for spatially varying relationships between dependent and independent variables [51]. The myth of the GWR model is the establishment of a spatial weight matrix defining a fixed or adaptive bandwidth used for borrowing data from surrounding sample points to obtain the particular estimates for any given sample point; thus, the locally weighted least square (WLS) method can be applied to calculate the varying estimates for samples across geographical units.

Multiscale geographically weighted regression (MGWR) was developed from geographically weighted regression (GWR), having better fitting and explanatory ability for spatial differences in variables than the original GWR model [52], and was here used to analyze the spatial heterogeneity of the effect of village geography on the willingness to resettle. GWR model assumes that all variables have the same action scale, which ignores the scale difference of variables in real situations, and MGWR model optimizes this. MGWR was able to compute the regression coefficients for each administrative village and was able to choose the appropriate bandwidth to calculate the regression coefficients according to the different scales at which the variables act [52]. We used the appropriate bandwidth to calculate the regression coefficients, and the calculation formula is as follows:

$$y_i = \sum_{j=1}^{k} \beta_{bwj}(u_i, v_i)x_{ij} + \varepsilon_i \tag{1}$$

where $x_{ij}$ is the value of variable $j$ for administrative village $i$, $bwj$ denotes the bandwidth used for the regression coefficient of variable $j$, $\beta_{bwj}$ is the regression coefficient of variable $j$, and ($u_i$, $v_i$) represents the spatial location of administrative village $i$. The MGWR calculation in this paper was based on the MGWR2.2 software developed by Arizona State University, where the bandwidth search mode was Golden Selection, and the spatial kernel function model was Bisquare.

## Results

### Descriptive statistics and spatial characteristics of WRV

**Descriptive statistics.**   There are 847,442 farm households in the study area, of which 453,356 households are willing to resettle, and the total willingness rate of the villages to resettle is 53.50%. Only 7.31% of the villages have a willingness rate of less than 10%, while 18.74%

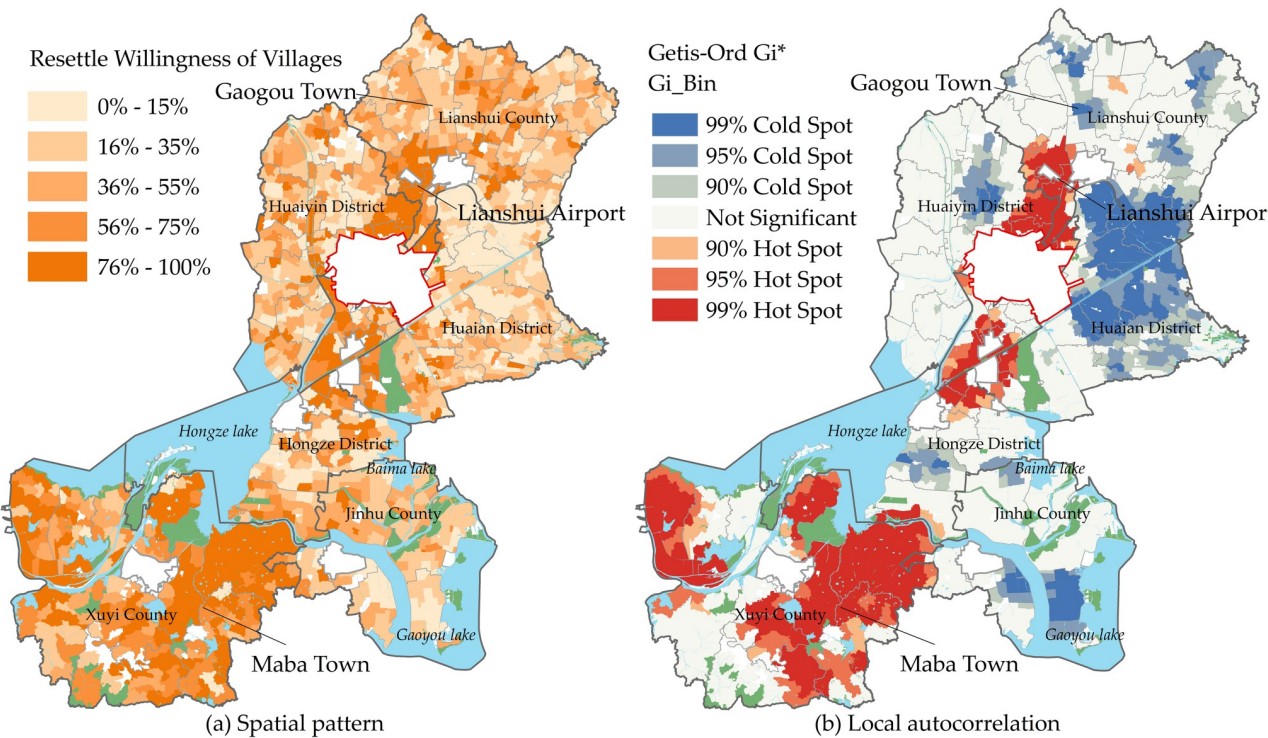

**Fig 3. (a) Spatial distribution of WRV; (b) Getis-Ord Gi\*map of WRV.**

have a willingness rate of more than 90%, and more than half of the villages have a willingness rate of 50%, which shows that there is an unavoidable trend for farmers to resettle in rural areas of Huai'an City and for rural settlements to be restructured in the future. The average willingness to resettle rate of the 1382 administrative villages in the study area is 41.64%, the maximum willingness rate is 100%, the minimum is 0%, the standard deviation is 0.30, and the coefficient of variation (CV) is 0.71. It can be seen that although the average willingness rate is high, the inter-village variation of willingness to resettle is large, and the spatial variation is obvious.

**Spatial distribution characteristics of WRV.** In terms of the spatial distribution of WRV, villages with high values are relatively concentrated in the "J"-shaped area extending from the northeast to the southwest of the study area (Fig 3a), with villages around Lianshui County in the northeast, Huai'an Lianshui Airport, Huai'an City, Hongze District in the middle, and villages in the whole territory of Xuyi County in the southwest having high WRV, which is basically consistent with the urban development axis of the study area. The global Moran's *I* of WRV is 0.38, with a z-score of 30.17 (greater than the critical value of 2.56), and it passes the significance test, indicating that WRV in the study area has a significant spatial correlation and is positively correlated. The results of the local spatial autocorrelation analysis are shown in Fig 3b. The cold spot areas are mainly concentrated in Huai'an District on the east side of the city, and the other cold spot areas are dispersed around the townships of Gaogou, Hongyao, Wugang and Tangji in Lianshui County, as well as the townships around the Hongze Lake, the Baima Lake and the Gaoyou Lake. The hotspots are distributed in a "ring" around the Xuyi urban area, industrial park, and Lianshui Airport. The spatial dispersion pattern of hot and cold spots of WRV is not the same, so it is difficult to analyze the factors influencing the

willingness to resettle overall and thus to obtain a real conclusion, which is also not conducive to the development of differentiated decision-making according to individual local conditions.

## Analysis of factors influencing WRV by village environmental elements

**Analytical model selection.** There is a strong spatial autocorrelation of WRV in the study area, and the traditional regression model does not consider this spatial feature, and so according to the above selected indicators of village environmental factors as the influencing factors, set the coordinates of the geometric center of the administrative village as $(u_i, v_i)$, and WRV as $y_i$, and construct the OLS, GWR, and MGWR models (Table 2).

Following the GWR roadmap proposed by Comber et al. (2020) [52, 53], an OLS model providing global benchmarking results was first run to preliminarily verify that the four elements of the village environment constructed in the previous section have a significant causal correlation with WRV. Compared to the OLS model, the GWR and MGWR models have the

**Table 2. Selection of variable for the analytical model.**

|  | (1) OLS | (2) GWR | (3) MGWR | (4) OLS | (5) GWR | (6) MGWR |
|---|---|---|---|---|---|---|
|  | $\beta$ | $\bar{\beta}$ | $\bar{\beta}$ | $\beta$ | $\bar{\beta}$ | $\bar{\beta}$ |
| Elements of Natural Environment |  |  |  |  |  |  |
| Water ($X_1$) | 0.094*** | 0.059 | 0.137 | 0.094*** | 0.070 | 0.198 |
| Terrain ($X_2$) | 0.085*** | 0.037 | 0.215 | 0.083*** | 0.009 | -0.009 |
| A11Y-City ($X_3$) | -0.031 | -0.205 | -0.372** | -0.033 | -0.116 | -0.291** |
| A11Y-County ($X_4$) | -0.130*** | 0.101 | 0.018 | -0.134*** | 0.024 | -0.016 |
| Farmland ($X_5$) | 0.030 | 0.066 | 0.042** | 0.033 | 0.118 | 0.023 |
| Homestead ($X_6$) | 0.135*** | 0.013 | 0.073*** | 0.139*** | 0.019 | 0.064** |
| Elements of Economic Environment |  |  |  |  |  |  |
| Revenue ($X_7$) | -0.034 | 0.001 | -0.022 |  |  |  |
| Building ($X_8$) | -0.268*** | -0.313** | -0.301*** | -0.300*** | -0.312* | -0.307*** |
| Housing ($X_9$) | -0.040 | -0.019 | -0.013 |  |  |  |
| GDP ($X_{10}$) | 0.062** | -0.002 | 0.043 | 0.063*** | -0.001 | 0.034 |
| Industry ($X_{11}$) | -0.024 | -0.043 | -0.020 |  |  |  |
| Network ($X_{12}$) | 0.017 | -0.006 | -0.007 | 0.010 | -0.007 | -0.014 |
| External ($X_{13}$) | -0.018 | 0.021 | 0.006 |  |  |  |
| Elements of Social Environment |  |  |  |  |  |  |
| Decentralized ($X_{14}$) | 0.063** | 0.028 | 0.027 | 0.061** | 0.022 | 0.033 |
| Density ($X_{15}$) | -0.005 | 0.004 | 0.023 |  |  |  |
| Household-Pop ($X_{16}$) | 0.212*** | 0.119 | 0.089*** | 0.209*** | 0.101 | 0.080 |
| High-school ($X_{17}$) | -0.135*** | -0.266 | -0.093 | -0.120*** | -0.168 | -0.104* |
| A11Y-Town ($X_{18}$) | 0.039 | -0.016 | -0.024 |  |  |  |
| Vacant ($X_{19}$) | 0.207*** | 0.125 | 0.132*** | 0.213*** | 0.124 | 0.129*** |
| Homebuyer ($X_{20}$) | -0.023 | 0.025 | 0.031 | -0.041 | 0.020 | 0.016 |
| Elements of Policy Environment |  |  |  |  |  |  |
| Restricted ($X_{21}$) | 0.103*** | 0.062 | 0.018 | 0.106*** | 0.056 | 0.020 |
| Improve ($X_{22}$) | -0.130*** | 0.074 | -0.120* | -0.129*** | -0.133 | -0.122* |
| Intercept ($P \leq 5\%$) | 0.000 | -0.284 | -0.233 | 0.000 | -0.204 | -0.307 |
| $R^2$ | 0.307 | 0.581 | 0.621 | 0.303 | 0.585 | 0.613 |
| Adj. $R^2$ | 0.296 | 0.507 | 0.569 | 0.295 | 0.517 | 0.564 |
| AICc | 3464.060 | 3208.754 | 2963.489 | 3459.920 | 3165.464 | 2967.093 |
| Log-likelihood | -1707.588 | -1359.674 | -1290.957 | -1711.709 | -1353.039 | -1304.276 |

highest $R^2$, Adj. $R^2$, and log-likelihood values and the lowest AICc values. Compared to GWR, MGWR has a higher goodness of fit and a significantly higher number of villages of significance, indicating that the results obtained after relaxing the assumptions of the modeling process at the same spatial scale are better in terms of goodness of fit and analytical results, suggesting that the MGWR model is better suited to explaining spatial variations of the influences than the other two models.

In order to verify the stability of the model, we first inserted all the variables of the village environmental elements into the three analytical models of OLS, GWR, and MGWR, respectively, and found that Water, Terrain, A11Y-city, A11Y-County, Farmland, Homestead, Building, GDP, Decentralized, Household-Pop, High-school, Vacant, Restricted, and Improve were more significant on both the global and the local scales, while the Network and Homebuyer factors were more locally significant. The 15 variables with significance were then brought back into the three analytical models 4–6, where the significance of the variables remained consistent between the simplified model and the original model, as did the ordering relationships between $R^2$, Adj. $R^2$, Log-likelihood, and AICc. Despite the reduction in the number of variables, the model results remain robust, indicating the reliability of the model specification and variable selection.

**Determinants.** In Table 2, the traditional OLS (Models 1 and 4) initially clarifies the determinants in the village environmental elements from a global perspective. In terms of the positivity and negativity of the variables, Water, Terrain, Homestead, GDP, Decentralized, Household-Pop, Vacant, and Restricted positively affect the rate of willingness to resettle in villages; that is, the higher the value of these variables, the higher the WRV. For example, the higher the vacancy rate of houses in the village, the stronger the willingness of farmers to resettle in the village. In addition, four variables, A11Y-County, Building, High-school, and Improve, negatively affect the willingness to resettle. In terms of the influencing power of the variables, the influence of the Building, House-hold-Pop, Vacant, Homestead, High-school, Improve, A11Y-County, Restricted, Water, Terrain, Decentralized, and GDP variables decreases in order. It can be seen that the variables related to housing, population, and land conditions, which are directly related to the improvement of the living environment of farm households, have the strongest influence. In terms of the significance of the variables, there are 10 variables with poor significance, of which the village economic environment has the most, and combining the influence of the previous variables, it can be found that the factors of the village economic environment affecting WRV are mainly focused on the housing conditions of families. With the inclusion of spatial location effects, the number of variables that were significant overall globally was significantly reduced, with only the Building variable being significant in the GWR model and the A11Y-City, Homestead, Building, Household-Pop, High school, Vacant, and Improve variables being significant in the MGWR model. In particular, the A11Y-City variable went from being insignificant and weakly influential in the OLS model to being significant and influential.

On a local scale, the number of villages of significance and the distribution of regression coefficients estimated using the MGWR model (Model 6) are presented in Fig 4. Of the 16 variables, only Terrain is not significant at all in all villages, while five variables, A11Y-City, Homestead, Building, Household-Pop, and Vacant, are significant in almost all villages, and the other variables are significant in localized villages. In terms of the degree of dispersion of the regression coefficients, five variables, Water, A11Y-City, A11Y-County, GDP, and Homebuyer, have the largest degree of dispersion, with large spatial heterogeneity, and four variables, Network, Decentralized, Restricted, and Improve, have a certain degree of dispersion. The remaining variables are less discrete. In terms of the magnitude of the local regression coefficients, six variables, A11Y-City, Building, GDP, Vacant, Homebuyer, and Improve, have a

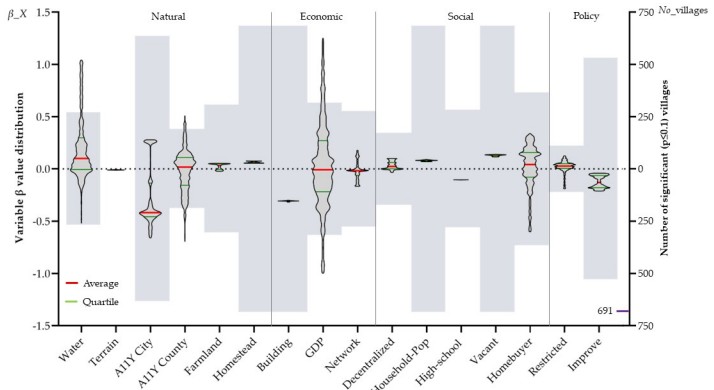

**Fig 4. Distribution of no. of significant villages and regression coefficients estimated by MGWR model.**

strong influence on some villages. In terms of positive and negative local influence, nine variables, namely Water, A11Y-City, A11Y-County, Farmland, GDP, Network, Decentralized, Homebuyer, and Restricted, have a bi-directional influence on WRV in different regions.

The combined global and local scale analysis shows that indicators affecting WRV are present in all four environmental elements of the village, with the natural environment element having the most variables. The ranking of the influence of village environmental elements is natural environment > economic environment > social environment > policy environment. Meanwhile, it can be seen that the geographic location factor is the key to the natural environment element, the housing condition is the key to the economic environment element, the trend of farmers' behavior is the key to the social environment element, and the planning guidance factor is the key to the policy environment element. A11Y-City, Building, Vacant, and Improve are the core variables of the above four factors, respectively. At the same time, there is obvious spatial heterogeneity in the village environmental elements, and there is a large gap between the scales of action of different variables.

**Scale effects of village environmental elements.** The MGWR model calculates different bandwidths for each variable, and the introduction of different bandwidths improves the explanatory power of the model and provides a basis for "scale effects" on the variables [54]. For example, the Water and A11Y-County bandwidths are 107 and 82, showing that for any given village, the effect that the village's water network density and time to the county town have on the willingness to resettle is stabilized in the nearest 100 or so villages to the county town. According to the variable bandwidths in Fig 5, the four variables Water, A11Y-County, GDP, and Homebuyer have the smallest bandwidths, have large spatial heterogeneity, and are close to the scale of towns in the study area, suggesting that the scope of the influence of these variables exists only at a certain town scale. The bandwidths of the variables A11Y-City, Network, and Policy Environment are 358, 428, 312, and 532, respectively, which are close to the number of county-level villages in the study area, suggesting that there is a sub-county (district) scale effect of these variables on willingness to resettle. The bandwidths of the remaining variables are basically close to the global level, indicating that the effects of these variables on willingness to resettle constitute a generalized phenomenon in the study area. Overall, most of the variables belonging to the natural and policy environment factors have a small impact scale, and most of the variables belonging to the economic and social environment factors have an impact scale close to the global scale, suggesting that the economic and social environment have a generalized effect on the willingness of farmers to resettle in the rural areas, and that there is a large spatial heterogeneity in the effects of the natural and policy environments.

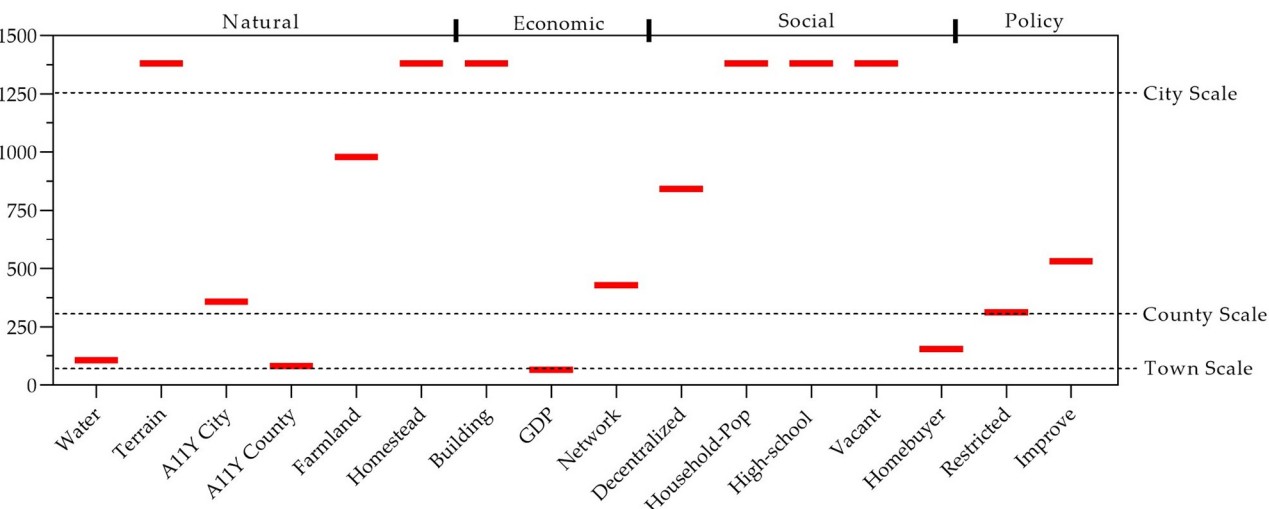

**Fig 5. Bandwidth of variable by MGWR model.**

## Mechanisms of village environmental elements on WRV

In addition to clarifying the determinants and scales of influence that affect the urban home purchasing of farm households both globally and locally, we also explore the internal mechanisms of the spatial variation effects of the determinants. On the basis of the theoretical analytical framework constructed in the previous section, we explain the villages' natural environment elements, economic environment elements, social environment elements, and policy environment elements, respectively. According to the calculation results of MGWR 2.2 software for the MGWR model, the local regression coefficients of 15 deterministic variables (the Terrain variable is not significant in all villages, so it will not be analyzed) are visualized, and the classification method is positive and negative stratification followed by grading in a way close to the natural discontinuity point, and the results are shown in Fig 6.

**The influence effect of natural environmental elements.** Most of the elemental variables of the three factors of the village's natural environment have a significant effect on the willingness to resettle, with the Natural Endowments and Geographical Location factors having a greater influence, and the Land Resources factor has a lesser influence. The Natural Endowments factor is sensitive to the willingness to resettle in the region. The Natural Endowments factor has patchy distribution, with more than 90% of villages being positively affected, especially in the western part of Lianshui County, the northern part of Huaiyin District, and the eastern part of Huai'an District, and shows an inner-high and outer-low circle, while the negatively affected areas are mainly located close to the northern part of the urban area and around the city of Xuyi County. The Geographical Location factor of A11Y-City has a significantly higher effect on willingness to resettle than A11Y-County. A11Y-City shows opposite effects in different regions, with a positive effect in Xuyi County in the south, a negative correlation in other regions, and a decreasing circle centered on Huai'an City. A11Y-County also affects willingness to resettle in a bidirectional way, but the regions with a larger effect are relatively few and scattered. The negatively affected areas are mainly distributed around the two counties in the north and south and the central industrial concentration area, Baima Lake Tourism Development Zone. The influence of the Land Resources factor is generally low and shows a spatial pattern of a gradient decrease from south to north.

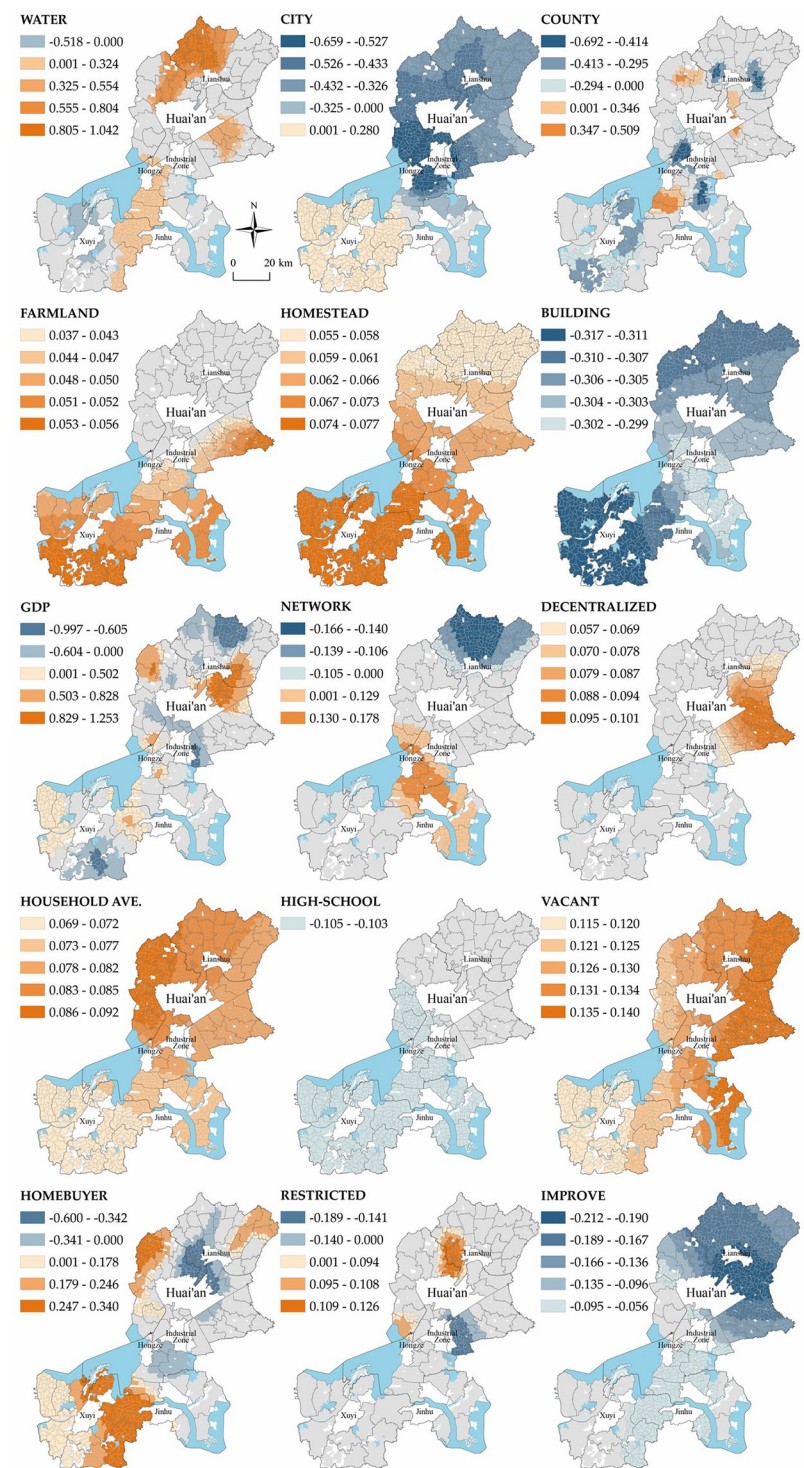

**Fig 6. Spatially varying estimates of variables.**

First of all, the area under study is a agricultural plains area, and the natural endowments of villages are mainly reflected in the density of the water network; however, water resources will have different impacts on villages in different locations. For villages in relatively remote locations, high water network density leads to lower road network density in the village, which affects the convenience of farmers' lives and thus promotes farmers' willingness to resettle. However, villages like Xuyi County and the outskirts of urban areas have relatively good transportation and living facilities, and villages with a high water network density have a better natural environment and can rely on the water to develop suburban rural tourism and increase employment and income, so the willingness of farmers to resettle will be weakened. Secondly, the closer the village is to the city, the less willing farmers are to resettle. On the one hand, this is because farmers in the periphery of the city can obtain public service spillover from the city, and at the same time can enjoy the beautiful natural environment of the countryside; on the other hand, with the expansion of the city, farmers in villages in the periphery of the city are likely to receive a large amount of money for demolition and resettlement. Meanwhile, the effect of urban spillover public services and the standards of demolition and resettlement are significantly higher than those of the county. However, counties far from urban areas are bound to be less attractive due to poorer transportation accessibility and the attraction of higher-level cities outside the urban area; thus, there is a higher willingness to resettle among farm households in villages further away from urban areas. Finally, the southern part of the study area is hillier and has more water resources, with poor land resources and a smaller average household size, so any small change in the size of arable land and residential land will affect the production and income of farmers, thus affecting their willingness to resettle.

**The influence effect of economic environmental elements.** The influence of village economic environment elements on the willingness to resettle is mainly realized through the factors of Income and Housing Conditions and Industrial Base, while the effect of the factor of Elemental Circulation Opportunities is smaller in terms of both effect and region. In particular, the strongest effect of the Income and Housing Conditions factor is found for Building, which shows a negative correlation, and the spatial pattern of gradient change is slightly weakened from the north and south inwards. The GDP variable of the host town is the core influencing factor of the village's Industrial Base factor, which is a significant spatial variability variable and affects the willingness of farmers to resettle in both directions, with the positive and negative influences staggered and decentralized. In the Elemental Circulation Opportunities factor, only the network variable affects farmers' willingness to resettle in both directions, and the positively affected areas are mainly distributed in Hongze District and Jinhu County in the central part of the country, while the negatively affected areas are mainly distributed in Lianshui County in the northern part of the country, with the effects showing a spatial pattern of circles.

The initial purpose of farmers' willingness to resettle is to seek a better living situation and better employment opportunities. First of all, the construction of buildings is not only representative of the economic strength of farmers, but also a manifestation of the improvement of their living environment. At the same time, there are a large number of farmers in rural areas of China who have already completed the relocation from townships to cities and have strong economic strength and a "land plot," and the construction of buildings in the countryside is a kind of status symbol, which is caused by a number of factors. The higher the proportion of buildings, the more reluctant farmers are to resettle. At the same time, the spatial characteristics of the distribution of influence reflect that the further away from the city, the higher the sensitivity to housing conditions. Secondly, the higher the financial income of townships (townships) with agriculture as the main industry, the higher the willingness of farmers to resettle, and the lower the willingness to resettle of farmers in areas with specialty industries

(tourism, manufacturing, etc.). Finally, in the central part of the study area, the overall level of the equivalent road network is low after being affected by the water network, but the degree of external connection is high, and the circulation of factors in the region creates a situation of low internal and high external pressure, which promotes the idea of resettling to more farmers, while Lianshui County in the north displays the opposite trend.

**The influence effect of social environmental elements.** The spatial fluctuations in the influence of the village social environment on willingness to resettle are less overall, with the Group Behavioral Trends factor having the strongest influence on willingness to resettle, and the Social Welfare and Population Concentration Degree factors at the village level having a weaker influence. Among them, Decentralized in the Population Concentration Degree factor positively influences willingness to resettle, and although the number of villages with significance is small, the spatial clustering is strong. The other variable, Household-Pop, although significant in all villages, has a weaker impact and less spatial fluctuation. The Social Welfare factor has only the High-school variable negatively influencing willingness to resettle in the southern part of the study area, and also has less spatial fluctuation. The Group Behavioral Trends factor has a positive impact on willingness to resettle. Vacant positively affects willingness to resettle, with a weakly increasing gradient from west to east and less spatial fluctuation in the effect. Only the Homebuyer variable has a large spatial difference in its effect and affects farmers' willingness to resettle in both directions, with the negatively affected areas concentrated around the urban area and gradually decreasing outward, and the positively affected areas mainly distributed away from the urban area, with the eastern part of Xuyi County, the northern part of Huaiyin District, and the eastern part of Lianshui County as the peak areas of the coefficients.

First, the dispersal of farm households in villages will lead to small and highly connected settlements, making it difficult for the government to invest in infrastructure development, making it difficult for farm households to obtain comprehensive public services and support, and making it easy for farm households to develop a willingness to resettle. This phenomenon is also verified at the farm household level, and the intensity of the impact is much more obvious than at the cluster level. This is because the dispersion of the farm household level is not only the dispersion of residences; increasing the rate of villages with high vacancy rates will lead to the destruction of the original social network ties of the cluster, and the degree of retention of farm households in the original social network cannot offset the desire to resettle, thus triggering a willingness to resettle. Second, because the overall household structure in the study area can be categorized into three types: one-person or two-person households with elderly people living alone, two-person or three-person households with two generations living together, and three-person or four-person or more households with three generations living together, an increase in the average household size of the villages connotes an increase in the proportion of middle-aged and young people, who choose to resettle in order to attain better employment and education. In addition, villages around urban areas have higher levels of development, better living conditions, and more room for value added near the urban areas, so farmers who have houses in towns are more inclined to keep their houses in villages, either as assets for investment or for retirement housing, while villages far away from the urban areas display the opposite trend.

**The influence effect of policy environmental elements.** The Restricted variable of the national Rigid policy element significantly affects the willingness to resettle in both directions, and the negatively affected areas are centrally distributed around the north shore of Lake Baima and the Beijing–Hangzhou Grand Canal, while the positively affected areas are mainly distributed around the north shore of Huai'an Airport and Hongze Lake. The Planning Policy factor variable Improve significantly affects the willingness to resettle in a negative direction,

and its effect is highest in the middle of Lianshui County and the urban area, and gradually decreases outward in a gradient, and, overall, the north side is higher than the south side.

National rigid policies have diametrically opposed effects in different regions, mainly because of the different factors of development restrictions. The dominant factors in development restriction areas that negatively affect the willingness to resettle are ecological protection zones, shipping lanes, and the proximity to water sources, which are areas where farmers can rely on natural resources for the development of rural tourism, and where farmers can find high-quality employment in the immediate vicinity, and thus have a low willingness to resettle. On the other hand, the dominant factors in these areas with a positive influence are airport construction restriction zones, high-speed rail line surroundings, chemical parks, etc. Farmers living in these areas are often affected by noise and air pollution, and the original social networks of villages are cut off. These dominant factors can easily become "neighbor avoidance facilities," and the high standard of resettlement payments in these areas leads to a high willingness to resettle. The spatial pattern influenced by planning-oriented policies is mainly due to the fact that the overall development level of villages decreases from the north to the south, especially in villages with a high density of farm households and a weak industrial base; these villages are highly sensitive to the government's planning-oriented policies. Villages with a high level of development have a high degree of contact with the outside world, and farmers have a high degree of autonomy, meaning that planning policies guided by local governments to resettle are not very attractive to the decision-making of farmers to resettle.

## Conclusion and policy recommendations

### Conclusion

Accurately measuring the relationship between village environmental factors and willingness to resettle can provide practical lessons for local governments' rural restructuring as well as urban–rural integration development, and also help to improve related theories, such as population geography. In this paper, we constructed a four-dimensional framework to analyze the environmental elements in villages, used spatial autocorrelation analysis, multiscale geographically weighted regression (MGWR) model, and other methods, and analyzed the data from a full-sample household survey of 1382 administrative villages in Huai'an City, a typical agricultural area in eastern China's plains, to characterize the spatial distribution of WRV, and to illustrate the determinants, scales of influence, and effect of the spatial heterogeneity of the environmental elements on WRV, and the conclusions are as follows.

In the study area, the willingness of farmers to resettle is high, and there are obvious differences in WRV among villages. The villages with a high willingness to resettle form a J-shaped spatial pattern extending from the northeast to the southwest, which is basically the same as the urban development axis in the study area, and the spatial dispersal of the high-value and low-value zones shows a completely different way. After comparing different models, it is found that the intensity of the influence (on decision-making) of the natural environment is the strongest among the village's environment elements, followed by the economic and social environments, and the weakest influence is found for the policy environment. The geographical location of the village is key to the natural environment element, the housing conditions of the villagers are key to the economic environment element, the behavioral trends of the farmers are key to the social environment, and the planning and guidance types of policies are key to the policy environment. It is also found that there is a significant difference in the scale of influence of village environmental elements on the willingness to resettle, with most of the variables in the natural and policy environment elements having a small degree of influence,

while most of the variables in the economic and social environment elements have a scale of influence close to the global scale.

Moreover, we find the inner mechanism of a willingness to resettle by analyzing the spatial variation effect of the factors affecting it. The specific results are as follows: natural endowment in the natural environment elements affects the willingness to resettle through the density of the village road network, the absolute location of the urban area in the geographic region has a significantly greater effect than that of the county and low spatial variation, and the effect of land resources is low. Farmers' income and housing conditions in the economic environment factor are the key to lowering the willingness to resettle, and the willingness to resettle is lower in relation to the Industrial Base factor in the areas with specialty industries, and the internal road facilities are the dominant factor in circulation opportunities. The degree of village agglomeration in the social environment element shows obvious stratification, with the highest influence at the family level and at the colony level, and the trend of group behavior is the dominant factor in the social environment element. Neighborhood avoidance facilities in the national rigid policy in the policy environment element trigger the resettling of farming groups, and villages with low levels of development are more likely to be affected by the planning guidance policies of the local government.

## Policy recommendations

The findings of this paper provide a scientific decision-making reference for the government to formulate differentiated policies, change the city-centered mindset, and innovate the village and township system. First, the spatial differences in the rate of willingness to resettle in villages and the effects of environmental factors in villages highlight the importance of the government to formulate different policies by region. The government should take the willingness of the farmers as one of the most important means to improve the management level of the resettlement service, understand the real willingness of the farmers through in-depth surveys, and formulate differentiated policies according to the spatial distribution of this willingness. Farmers in villages on the outskirts of urban centers are more willing to improve their living conditions through resettlement; at this time, we can make full use of the strong financial strength of the urban centers to accept the resettlement of retired farmers by giving enough monetary compensation, which can help them to quickly integrate into urban life. Remote rural areas should be treated according to the characteristics of the village environment. In rural areas with no special industries, the local government should comply with the will of the farmers to resettle, improve the public service facilities of the resettlement community, and at the same time, give full play to the advantages of the scarcity of land and agricultural production in the development of large-scale agricultural cultivation to ensure national food security. In rural areas where there are specialty industries, the government should further improve the living environment for farmers by improving public service facilities in villages.

Secondly, the government needs to change the unidirectional thinking and policy orientation that has so far relied excessively on the urbanization of rural populations through purchasing homes and settling in destination towns and cities. The government should take into account the diversified urban housing needs of both out-migrants and in-migrants in an integrated urban–rural approach, so as to realize a diversified urbanization path that combines relocation and localization.

Finally, there is a need to scientifically promote the planning and layout of villages and towns, and to actively cultivate sub-core towns and villages to give play to the centralizing effect. The key strong towns and villages at the grassroots level play a leading role in the economic and social development of a certain rural space, and give a strong impetus to rural

population agglomeration, public services, and industrial development. Actively fostering sub-township cores with better foundations within the county through planning, policy and other means can reduce the loss of local vitality, promote in situ urbanization in the countryside, and contribute to the effective revitalization of rural areas to a greater extent.

## Prospects for research

Due to the scale of analysis and the actual situation, the selected indicators cannot fully express the elements of the village environment; identification of the mechanisms of action is not accurate enough. In addition, because the generation of a willingness to resettle in farmers can occur randomly, different times and environments may cause very different opinions, which may affect the spatial distribution of farmers' will. The next step is to verify the real will of farmers through surveying individual farmers. In terms of theoretical research, the analytical framework and mechanisms of village environmental factors that are constructed here need to be empirically explored and verified in other types of regions.

## Supporting information

**S1 Table. Village names and related data.**
(XLS)

**S2 Table. Resettlement willingness of villages.**
(XLS)

## Author Contributions

**Conceptualization:** Chengxiang Wang, Pinrong He, Chang Gyu Choi.

**Data curation:** Chengxiang Wang, Pinrong He.

**Formal analysis:** Pinrong He.

**Funding acquisition:** Chengxiang Wang, Chang Gyu Choi.

**Investigation:** Pinrong He, Chang Gyu Choi.

**Methodology:** Chengxiang Wang.

**Project administration:** Chang Gyu Choi.

**Resources:** Chang Gyu Choi.

**Software:** Chengxiang Wang, Pinrong He.

**Supervision:** Chang Gyu Choi.

**Visualization:** Chengxiang Wang.

**Writing – original draft:** Chengxiang Wang, Pinrong He.

**Writing – review & editing:** Chang Gyu Choi.

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
