## [Decision Letter · Decision Letter 0]

4 Feb 2024

PONE-D-23-35877Resettlement willingness: From a village environmental perspectivePLOS ONE

Dear Dr. Wang,

Thank you for submitting your manuscript to PLOS ONE. After careful consideration, we feel that it has merit but does not fully meet PLOS ONE’s publication criteria as it currently stands. Therefore, we invite you to submit a revised version of the manuscript that addresses the points raised during the review process.

We look forward to receiving your revised manuscript.

Kind regards,

Madhulika Sahoo, Ph.D

Academic Editor

PLOS ONE

 [This research was supported by the National Natural Science Foundation of China (52078237); and the National Research Foundation of Korea grant, which is funded by the Government of South Korea (NRF-2020R1A2C1008509).].  

5. We note that Figure 2, 3 and 6 in your submission contain [map/satellite] images which may be copyrighted. All PLOS content is published under the Creative Commons Attribution License (CC BY 4.0), which means that the manuscript, images, and Supporting Information files will be freely available online, and any third party is permitted to access, download, copy, distribute, and use these materials in any way, even commercially, with proper attribution. For these reasons, we cannot publish previously copyrighted maps or satellite images created using proprietary data, such as Google software (Google Maps, Street View, and Earth). For more information, see our copyright guidelines: http://journals.plos.org/plosone/s/licenses-and-copyright.

a. You may seek permission from the original copyright holder of Figure 2, 3 and 6 to publish the content specifically under the CC BY 4.0 license.  

Reviewers' comments:

Reviewer's Responses to Questions

**Comments to the Author**

1. Is the manuscript technically sound, and do the data support the conclusions?

Reviewer #1: Yes

2. Has the statistical analysis been performed appropriately and rigorously? 

Reviewer #1: Yes

3. Have the authors made all data underlying the findings in their manuscript fully available?

Reviewer #1: Yes

4. Is the manuscript presented in an intelligible fashion and written in standard English?

Reviewer #1: Yes

5. Review Comments to the Author

Reviewer #1: Studying the willingness of rural residents to resettle is of practical significance for the implementation of the rural revitalization strategy. Unlike previous studies that analyzed individual differences through questionnaires, this study takes 1382 villages in Huai'an City, Jiangsu Province as the analysis unit. It investigates the factors affecting the overall willingness of resettlement in terms of natural, economic, social, and policy environment in rural areas, which is quite innovative. In particular, the author is able to examine the spatial heterogeneity of various influencing factors through methods such as geographically weighted regression (GWR) and Multiscale geographically weighted regression , and can provide good explanations for the research results, which are commendable. Finally, the author also puts forward policy suggestions based on the conclusions of this study, reflecting its practical value. Overall, this study has a clear thinking, smooth writing, specific and complete content, and is a well-completed paper. It is recommended for publication. Here, I only have one suggestion that I hope will be helpful for the author to improve the paper. That is, although the author explains the results based on the MGWR model in the paper, the explanation of this model is insufficient. For example, what is the difference between it and GWR, how to choose its bandwidth, etc. It is suggested to supplement these aspects so that readers can further understand this method.

6. PLOS authors have the option to publish the peer review history of their article (what does this mean?). If published, this will include your full peer review and any attached files.

Reviewer #1: No

---

## [Author Response · Author response to Decision Letter 0]

29 Feb 2024

Q1: When submitting your revision, we need you to address these additional requirements. Please ensure that your manuscript meets PLOS ONE's style requirements, including those for file naming.

Reply to: The following changes were made to the article in accordance with PLOS ONE style requirements:

1. Figures 2, 5 and 6 have been repositioned so that the title of each figure appears directly after the paragraph in which it is first cited.

2. Changed the position of Table 2 so that it appears directly after the first cited paragraph.

3. Added a section on "Supporting information" and provided two tables of supporting research data in this section, which have been named as requested.

Q2: Please note that PLOS ONE has specific guidelines on code sharing for submissions in which author-generated code underpins the findings in the manuscript. In these cases, all author-generated code must be made available without restrictions upon publication of the work.

Reply to: This article is not about CODE.

Q3: Thank you for stating the following financial disclosure: [This research was supported by the National Natural Science Foundation of China (52078237); 

Reply to: Project funders are corresponding and third authors of the manuscript, respectively, and play a leading role in study design, data collection and analysis, and decisions to publish or prepare the manuscript. We have included this revised funder role statement in the cover letter.

Q4: We note that your Data Availability Statement is currently as follows: [All relevant data are within the manuscript and its Supporting Information files.]

Reply to: A "Supporting information" section has been added to the manuscript and two tables of data supporting the study are provided in this section. These are S1 Table. and S2 Table.

Q5: We note that Figure 2, 3 and 6 in your submission contain [map/satellite] images which may be copyrighted.

Reply to: The administrative boundary data and elevation data are acquired from the Resources and Environment Science and Data Center of the Chinese Academy of Sciences (www.resdc.cn).

Q6: Please review your reference list to ensure that it is complete and correct.

Reply to: References were checked and articles that had been retracted or were difficult to search were removed and replaced with relevant references.

Comments to the Author

Q1: That is, although the author explains the results based on the MGWR model in the paper, the explanation of this model is insufficient. For example, what is the difference between it and GWR, how to choose its bandwidth, etc. It is suggested to supplement these aspects so that readers can further understand this method.

Reply to: The main difference between MGWR and GWR lies in the calculation of the scale of influence of the variables. GWR calculates the regression model based on the same scale of influence of the variables, which ignores the differences in the scale of the variables, while MGWR optimizes this aspect by calculating the scale of influence of each variable in the process of calculating the regression model. The revised version of the manuscript includes a description in the section "Overview of the study area and data sources".

Latest response: The administrative boundary data and elevation data are from the Center for Resource and Environmental Sciences and Data of the Chinese Academy of Sciences (www.resdc.cn). According to the Resource and Environmental Sciences Data Registration and Publication System (https://www.resdc.cn/DOI/) data users can cite the results of this data as if it were a paper. The revised manuscript has cited the data as the 50th reference of the paper.

At the same time, we found that some of the papers published by PLOS one are also cited in this way. For example, "Spatial distribution and influencing factors of high-quality tourist attractions in Shandong Province, China" https ://doi.org/10.1371/journal.pone.0288472

---

## [Decision Letter · Decision Letter 1]

31 May 2024

Resettlement willingness: From a village environmental perspective

PONE-D-23-35877R1

Dear Dr. Wang,

We’re pleased to inform you that your manuscript has been judged scientifically suitable for publication and will be formally accepted for publication once it meets all outstanding technical requirements.

Kind regards,

Biswajit Pal, M.SC., Ph.D

Academic Editor

PLOS ONE

Additional Editor Comments (optional):

Reviewers' comments:

Reviewer's Responses to Questions

**Comments to the Author**

1. If the authors have adequately addressed your comments raised in a previous round of review and you feel that this manuscript is now acceptable for publication, you may indicate that here to bypass the “Comments to the Author” section, enter your conflict of interest statement in the “Confidential to Editor” section, and submit your "Accept" recommendation.

Reviewer #1: All comments have been addressed

Reviewer #2: All comments have been addressed

2. Is the manuscript technically sound, and do the data support the conclusions?

Reviewer #1: Yes

Reviewer #2: Yes

3. Has the statistical analysis been performed appropriately and rigorously? 

Reviewer #1: Yes

Reviewer #2: Yes

4. Have the authors made all data underlying the findings in their manuscript fully available?

Reviewer #1: Yes

Reviewer #2: Yes

5. Is the manuscript presented in an intelligible fashion and written in standard English?

Reviewer #1: Yes

Reviewer #2: Yes

6. Review Comments to the Author

Reviewer #1: (No Response)

Reviewer #2: See attachment

Accept with minor revision

Reviewer’s report

Resettlement willingness: From a village environmental perspective

The study is interesting and is based upon a very contextual issue- resettlement willingness. It is particularly important in the context of rapid urbanization, transforming rural spaces and communities and nuanced rural urban interactions. The paper is relevant thematically and the conclusions are substantial. However, the following issues must be addressed:

1.Theoretical framework: The discussion under this section may better be called “conceptual framework”. The discussion is repetitive. The author may consider adding a table indicating factors/element that encourage willingness to resettle and those that discourage the same. This section must be revised and condensed.

2.Policy as an explanatory variable useful. However, on p. 18 the discussion section must refer to the existing policies clearly to strengthen the quantitative results. This aspect in the present form is weak and must be strengthened.

3.The discussions too elaborate and slightly repetitive resulting in a paper of more than 11000 words. It must be shortened to around 8000 words preferably.

4.The paper must be structured as follows: (1) Introduction, (2) Conceptual framework, (3) Study area (4) Materials and Methods (5) Results and discussion. In the present form the paper looks a little unstructured with the “materials and methods” section getting mixed up with the results.

5.The last section “Prospects of Research” appears like a statement outlining the limitations of the work. In that case it is better to move this section to the last paragraph of the “materials and methods” section.

I recommend “accept with minor revisions”.

7. PLOS authors have the option to publish the peer review history of their article (what does this mean?). If published, this will include your full peer review and any attached files.

Reviewer #1: No

Reviewer #2: No

---

## [Editor Report · Acceptance letter]

7 Jun 2024

PONE-D-23-35877R1 

PLOS ONE

Dear Dr. Wang, 

I'm pleased to inform you that your manuscript has been deemed suitable for publication in PLOS ONE. Congratulations! Your manuscript is now being handed over to our production team.

Kind regards, 

on behalf of

Dr. Biswajit Pal 

Academic Editor

PLOS ONE